# Women with HPV-Driven Anal and Genital Disease: Investigating the Patient Cohort in England

**DOI:** 10.3390/cancers17243970

**Published:** 2025-12-12

**Authors:** Micol Lupi, Sofia Tsokani, Ann-Marie Howell, Paris Tekkis, Christos Kontovounisios, Irene Chong, Sarah Mills

**Affiliations:** 1Department of Surgery and Cancer, South Kensington Campus, Imperial College London, London SW7 2AZ, UK; p.tekkis@imperial.ac.uk (P.T.); c.kontovounisios@imperial.ac.uk (C.K.); sarah.mills58@nhs.net (S.M.); 2Department of Colorectal Surgery, Chelsea and Westminster NHS Foundation Trust, 369 Fulham Road, London SW10 9NH, UK; ann-mariehowell2@nhs.net; 3Laboratory of Hygiene, Social & Preventive Medicine and Medical Statistics, School of Medicine, Aristotle University of Thaessaloniki, 541 24 Thessaloniki, Greece; stsokani@auth.gr; 4Department of Colorectal Surgery and Cancer, The Royal Marsden NHS Foundation Trust, 203 Fulham Road, London SW3 6JJ, UK; irene.chong@rmh.nhs.uk; 52nd Surgical Department, HYGEIA Hospital, Erithrou Stavrou 4, Marousi, 151 23 Athens, Greece; 6Department of Surgery, NYU Grossman School of Medicine, 550 1st Avenue, New York, NY 10016, USA; 7School of Medicine, National and Kapodistrian University of Athens, 157 72 Athens, Greece

**Keywords:** anogenital cancer, anogenital HSIL, women, HPV

## Abstract

Anal cancer is more common in women with a background of genital precancerous and cancerous lesions. Despite this, very little research has focused on this high-risk population. This study aims to investigate this population to better understand their risk factors for anal cancer and the relationships between genital and anal pathology. These findings add to the body of evidence demonstrating the risk of anal cancer in women who had cervical, vaginal or vulval pathology. They show that these women develop anal cancer at a younger age than women who have never had genital pathology and strengthen the argument to consider anal cancer screening for these patients.

## 1. Introduction

Anal squamous cell carcinoma (SCC) is a high-risk Human Papillomavirus (hrHPV)-induced malignancy [1,2]. People with recurrent and persistent exposure to oncogenic high-risk strains of HPV (16, 18, 31, 33, 35, 39, 45, 51, 52, 56, 58 and 59) are at higher risk of developing neoplastic anogenital lesions; in the context of anal cancer, HPV16 is thought to be the most carcinogenic [3]. The incidence of anal cancer has concerningly been on the rise [4,5], and whilst, relatively rare in the general population, affecting 1–2 per 100,000 people [6], there are populations who are at higher risk. These include people living with HIV (PLWH) (especially men who have sex with men (MSM)), transplant patients and women with other genital HPV-driven dysplasias [7]. These populations have been shown to have incidences between 22 and 85 (PLWH), 13 (transplant) and 9–48 (women with genital dysplasia) per 100,000 person-years, respectively [8].

Like other genital HPV-driven disease, anal SCC arises from a precancerous precursor called high-grade squamous intraepithelial lesion (HSIL). The ANCHOR trial published in 2022 was the first study to demonstrate that treating anal HSIL effectively prevents its progression to cancer [9], and as result of this, the International Anal Neoplasia Society (IANS) has published the first recommendations for anal cancer screening, which recognise the need to screen these high-risk populations [10]. However, these guidelines are largely based on expert consensus, given the limited published evidence on anal cancer, which has to this day focused on PLWH, especially MSM. This is despite the fact that two-thirds of all anal cancer diagnoses are in women [5], and few of these women (3%) have HIV [11]. Importantly, the incidence of anal cancer has been rising faster for women compared to men, especially in the 55+ years age group [5,12], where the incidence of anal cancer has now surpassed that of cervical cancer [13,14]. Concerningly, women have also been shown to be more likely to present with advanced staging, to receive radiotherapy and to die of anal cancer than men [15,16].

Persistent genital hrHPV infection and the resultant genital HSIL and/or SCC are now acknowledged independent risk factors for anal HSIL and SCC in women [8,10]. The perineum acts as a reservoir for hrHPV, which can readily spread around the anogenital region, eventually causing disease in women who are unable to clear it [17].

To our knowledge, this is the first study to look at women with both anal and genital HPV-related disease in England on a national level. It aims to better understand their sociodemographic risk factors, as well as timelines between acquisition of genital and anal pathology. These results will be important in ensuring that current anal cancer screening recommendations in this population are appropriate and in influencing potential evidence-based changes, where evidence is currently lacking.

## 2. Materials and Methods

This is a retrospective cross-sectional study which follows the ‘strengthening the reporting of observational studies in epidemiology’ (STROBE) statement [18]. The study was sponsored by the Imperial College London, received ethics approval by the London-Surrey Research Ethics Committee (IRAS ID 315094, REC reference 22/PR/0768) and was funded by the Red Trouser Day Charity, the National Institute for Health and Care Research (NIHR) Biomedical Research Centre at The Royal Marsden NHS Foundation Trust and The Institute of Cancer Research, The Syncona Foundation and The Royal Marsden Cancer Charity.

The dataset was requested through the NHS Digital data access request service (DARS). The application requested pseudonymised national cancer registration analysis service (NCRAS) data on all women over the age of 25 years, diagnosed with anal cancer and/or HSIL between 2001 and 2020, who also had a vulval and/or vaginal and/or cervical cancer and/or HSIL diagnosis within the 20-year period before, or 1-year period after, their anal cancer/HSIL, respectively. Data categories included age at diagnosis (5-year age intervals), month and year of diagnosis, ethnicity, deprivation, cancer staging, Charlson co-morbidity index (CCI), route to diagnosis and treatment received for the condition. Ethnicity groups were based on the 16 + 1 ethnic data categories defined in the 2001 census, which is the national mandatory standard [19]. It is important to note that the Unknown ethnicity category, as defined by the census, includes ‘Chinese’, ‘any other ethnic group’, ‘not stated’ and ‘not known’, and for this reason was included in all analysis. Deprivation scores were based on the Index of Multiple Deprivation (IMD), which is the official measure of deprivation in England [20], and are expressed in quintiles, with quintile 1 being the most deprived and quintile 5 being the least deprived. Staging for anal cancer was based on the 8th edition of the Union for International Cancer Control TNM (tumour, nodes, metastasis) Classification of Malignant Tumours (UICC TNM8) [19,21]. Cervical, vaginal and vulval cancer staging was based on the International Federation of Gynaecology (FIGO) classification FIGO 2018, 2009 and 2021, respectively [19]. The CCI is a measure of comorbidity severity as follows: mild (1–2), moderate (3–4) and severe (5) [22]. Routes to diagnosis were classified into emergency, GP referral, inpatient, other outpatient, screening, two-week wait (TWW) and unknown. Treatments were classified into chemoradiotherapy (CRT), cryotherapy, laser therapy, radiotherapy, surgery, chemotherapy, brachytherapy and unknown.

The dataset was registered with the Imperial College London Dataset Asset Registration Tool (DART) (ID 1285), and a Data Protection Impact Assessment (DPIA) was completed, assessed and approved by the College’s Information Governance and Data Protection teams. The final dataset was stored and analysed in an ISO 27001:2013 [23] certified secure environment provided by the Big Data Analytical Unit (BDAU) at Imperial College London.

Data analysis was conducted using R V4.2.3, Stata 18, Microsoft Excel V16.94 and GraphPad Prism V10.0.3. Frequency tables, bar charts and line plots were created to investigate patterns and relationships between different data categories. Frequency tables were used to explore the distribution of the categorical variables, whilst bar charts and line plots were used to visually demonstrate distributions.

A multivariate multinomial logistic regression analysis was conducted to examine the associations between ethnicity, deprivation and anogenital cancer staging, as well as ethnicity, deprivation and route to the diagnosis of anogenital disease. Similarly, a multivariate multinomial logistic regression analysis was conducted to determine predictors of anal cancer staging, as well as the number of anogenital diagnoses. Results were summarised using odds ratios (OR), alongside 95% confidence intervals (CI) and *p*-values. Additionally, we employed grouped bar plots to illustrate the distribution of the different factors explored per analysis and explore analysis issues. All analyses were conducted in R version 4.5.0, using the ‘nnet’ R-package for multinomial regression analysis and ‘ggplot2’ for data visualisation. A significance level of 5% was used.

For multivariate analysis, deprivation was grouped into high (quintiles 1–2) and low (quintiles 4–5), deprivation quintile 3 was excluded from the analysis. Staging was grouped in HSIL, early (stage 1–2) and late (stage 3–4). Number of diagnoses were grouped into 2–4 and 4–5. Age was grouped into <55 years and >55 years. Early staging, White ethnicity, high deprivation, emergency presentation, age < 55 years, 2–3 anogenital diagnoses, CCI 0, cervical HSIL and CRT were set as references for analysis. Age groups were also grouped into <40, 40–54, 55–74 and 75+ years in our descriptive data summaries. Except for ethnicity, where the ‘Unknown’ category was kept for analysis, the ‘not stated’, ‘unknown’ or ‘missing’ values were treated as missing.

## 3. Results

### 3.1. Pathologies

This cancer registry data request identified 1297 women diagnosed with anal disease who had a history of genital disease. A total of 2872 anogenital pathologies were diagnosed in this cohort, with 564 and 733 women diagnosed with anal HSIL or anal cancer, respectively. A total of 34 (5.8%) had both anal HSIL and cancer; these were grouped in the anal cancer category (see Table 1).

Furthermore, 5.6% (*n* = 72/1297), 41.9% (*n* = 543/1297), 1.5% (*n* = 19/1297), 4% (*n* = 52/1297), 21.2% (*n* = 275/1297) and 44% (*n* = 571/1297) of women also had a diagnosis of cervical cancer, cervical HSIL, vaginal cancer, vaginal HSIL, vulval cancer and vulval HSIL, respectively, in the 20-year period before or 1 year after their anal diagnosis (see Table 1).

In order to establish the proportion of patients with anal HSIL and/or cancer who also had a genital HSIL and/or cancer diagnosis on a national level, published NCRAS data [24] was downloaded to find the total number of female patients, 25 years or older, diagnosed with anal HSIL and cancer in England between 2001 and 2020. A total of 3646 and 12,655 women were diagnosed with anal HSIL or cancer, respectively. Of these, 15.5% (*n* = 564/3646) and 5.8% (*n* = 733/12,655) also had a genital pathology diagnosis. In total, 8% (*n* = 1297/16,301) of all women with a diagnosis of anal HSIL and/or cancer also had a genital pathology (see Table 1).

A total of 81.9% (*n* = 1062/1297) of patients had a diagnosis of anal HSIL and/or cancer and one other genital pathology (2 diagnoses in total), 15% (*n* = 194/1297) had 2 other anogenital pathologies (3 diagnoses in total), 3% (**n* =* 39/1297) had 3 other anogenital pathologies (4 diagnoses in total) and 0.2% (*n* = 2/1297) had 4 other anogenital pathologies (5 diagnoses in total).

Cervical and vulval HSIL were the most frequent first diagnoses at 39.8% (*n* = 516/1297) and 32.5% (*n* = 421/1297), respectively (see Table 2). The genital conditions were most frequently diagnosed as the first diagnoses.

### 3.2. Age

A total of 47% (*n* = 1350/2872) of pathologies were diagnosed in the <40 years age group (see Appendix A Appendix A), with 42.9% (*n* = 258/601), 50% (*n* = 36/72), 80.7% (*n* = 439/544), 61.5% (*n* = 32/52), 33.6% (*n* = 93/277) and 56.5% (*n* = 324/573) of anal HSIL, cervical cancer, cervical HSIL, vaginal HSIL, vulval cancer and vulval HSIL diagnoses having been made in this age group, respectively (see Appendix A Appendix A). Anal cancer and vaginal cancer diagnoses were most frequently made in the 55–74 years age group at 35.5% (*n* = 259/734) and 47.4% (*n* = 9/19), respectively.

### 3.3. Ethnicity, Deprivation and Route to Diagnosis

In total, 55.7% (*n* = 723/1297) of patients were from the deprivation 1 and 2 quintiles (see Table 3, 93.4% (*n* = 1211/1297) were of White ethnicity, and with the ‘Unknown’ route to diagnosis category removed, 51.7% (*n* = 847/1638) of pathologies were referred by the GP (see Table 3 and Table 4).

For Asian, Black and Unknown ethnicities, a large proportion of diagnoses were made in patients from the deprivation 1 quintile (100% (*n* = 1/1), 51.1% (*n* = 24/47) and 41.9% (*n* = 13/31), respectively. A total of 57.1% (*n* = 4/7) of Mixed patients were from the deprivation 2 quintile, and 54.9% (*n* = 665/1211) of White patients were from the deprivation 1 and 2 quintiles (see Table 3).

The most common routes to diagnoses were GP referral and other outpatient referral for all ethnicities, see Table 4. There were no two-week wait referrals in the Mixed ethnicity group, and the Black ethnicity had the highest proportion of screening detected anogenital pathology at 1.8% (*n* = 2/113) (see Table 4).

All Asian (*n* = 1) and Mixed ethnicity patients (*n* = 7) had in total 2 diagnoses, Unknown ethnicity patients had 2–3 diagnoses (*n* = 27 and *n* = 3, respectively), Black patients had up to 4 diagnoses (*n* = 32, *n* = 11 and *n* = 4, respectively), and White patients had up to 5 diagnoses, (*n* = 995, *n* = 179, *n* = 35 and *n* = 2, respectively) (see Table 4).

### 3.4. Staging and Treatment

A total of 38.3% (*n* = 136/355) of anal cancers were diagnosed at stage 3 disease; 65% (*n* = 13/20) of cervical cancers were diagnosed at stage 1 disease; 50% (*n* = 3/6) and 33.3% (*n* = 2/6) of vaginal cancers were diagnosed at stage 1 and stage 4 disease, respectively. A total of 86.8% (*n* = 66/76) of vulval cancers were diagnosed at stage 1 disease. HSIL was most abundant for all four anogenital areas (see Table 5).

Only 582 pathologies had a secondary treatment; this included treatments labelled as the ‘Unknown’ group (see Appendix A Appendix A). Surgery was the primary treatment modality for 71.6% (*n* = 2056/2872) of the pathologies. This is likely reflective of the fact that all conditions would have needed a surgical biopsy to confirm a histological diagnosis and plan definitive management. It is important to note that the intention of the surgical treatment was, however, not specified, nor was the indication for the secondary treatment, i.e., whether it was for definitive treatment, disease persistence or recurrence. Chemoradiotherapy (5.3%, *n* = 151/2872) and laser therapy (2.1%, *n* = 60/2872) were the second and third most common primary treatments (see Appendix A Appendix A). Chemoradiotherapy (43.1%, *n* = 251/582) and laser therapy (10.7%, *n* = 62/582) were the most common secondary treatments for the precancerous and cancerous pathologies, respectively (see Appendix A Appendix A).

### 3.5. Multivariate Logistic Regression Analysis

#### 3.5.1. Ethnicity, Deprivation and Anogenital Cancer Staging

There were no significant relationships between ethnicity and early vs. late staging, nor ethnicity and HSIL vs. early staging (see Appendix A Appendix A).

#### 3.5.2. Ethnicity, Deprivation and Route to Diagnosis of Anogenital Disease

Only one analysis, yielded interpretable and significant results; compared to women of White ethnicity, women of Black ethnicity were shown to be 92% (OR 0.08, 95% CI 0.01–0.7, *p* = 0.02) less likely to present via the two-week wait referral pathways than present as an emergency (see Appendix A Appendix A).

#### 3.5.3. Predictors of Anal Cancer Staging

There were no statistically significant relationships between deprivation, first route to diagnosis, number of diagnoses, route to diagnosis for anal pathology, Charlson Comorbidity Index at first diagnosis and anal cancer staging (late- vs. early-stage nor HSIL vs. early-stage cancer) (see Appendix A Appendix A).

With respect to ethnicity, Black ethnicity was the only group to produce interpretable results, which were of borderline significance; compared to White patients, Black patients were 864% (OR 9.64, 95% CI 0.74–126, *p* = 0.08) more likely to present with late-stage than early-stage anal cancer (see Appendix A Appendix A).

There was a significant relationship between age and HSIL vs. early anal cancer staging. Individuals aged 55 years and older were 80% (OR 0.20, 95% CI 0.08–0.55, *p* < 0.001) less likely to have an anal HSIL than early-stage anal cancer diagnosis compared to women aged <55 years of age (see Appendix A Appendix A).

Lastly, there were significant relationships between the site of the first anogenital diagnosis and anal cancer staging. Compared to women who were first diagnosed with cervical HSIL, those who were first diagnosed with vulval cancer were 89% (OR 0.11, 95% CI 0.02–0.68, *p* = 0.02) less likely to have a late-stage vs. early-stage anal cancer diagnosis; this was a significant finding (see Appendix A Appendix A). A similar finding of borderline statistical significance was seen for women who were first diagnosed with vulval HSIL (OR 0.21, 95% CI 0.05–1, *p* = 0.05), who were 79% less likely to have a late- vs. early-stage anal cancer diagnosis (see Appendix A Appendix A).

Women who were first diagnosed with vulval HSIL were 271% (OR 3.71, 95% CI 1.09–9.25, *p* = 0.03) more likely to have an anal HSIL vs. early-stage anal cancer diagnosis (see Appendix A Appendix A).

#### 3.5.4. Predictors of Number of Anogenital Diagnoses

Only one analysis yielded interpretable and significant results; compared to women who were first diagnosed with cervical HSIL, those who were first diagnosed with vulval HSIL were 87% less likely to have 2–3 vs. 4–5 genital diagnoses (OR 0.13, 95% CI 0.02–0.91, *p* = 0.04) (see Appendix A Appendix A).

#### 3.5.5. Timelines

Figure 1 and Figure 2 illustrate the average number of years between a genital pathology and an anal pathology diagnosis in this cohort. The number of years between cervical, vulval, vaginal and an anal diagnosis ranged between 0 and 26 years, 0–23 years and 0–20 years, respectively. The number of years between an anal HSIL and an anal cancer diagnosis ranged between 0 and 12 years.

Figure 3 summarises the average number of years between different diagnoses. The gap between the first and second diagnosis was the largest, with a median of 7 years, whilst the gaps between second and third, third and fourth and fourth and fifth gradually decreased with each consecutive diagnosis.

## 4. Discussion

Between 2001 and 2020, 1297 women diagnosed with anal HSIL and/or cancer also had synchronous or metachronous genital HSIL and/or cancer in England. On a national level, this amounted to 8% of all women with anal disease also having historical or active genital pathology, with vulval and cervical HSIL representing the most frequently diagnosed genital conditions.

Whilst, to our knowledge, there are no other publications studying this population on a national level, a study reviewing 15 years’ worth of anal cytology screening at the Mayo clinic reported 26.1% and 8.0% of anal HSIL patients to also have vulval and cervical HSIL [25]. These numbers are comparable to those in this study, which found 10.3% and 5.2% anal HSIL patients to also have vulval and cervical HSIL. Similarly, another study reported that 1.63% of women with anal cancer had a previous genital cancer diagnosis [26], which is comparable to the 1.9% found in this study.

In this population, most had two affected anogenital zones. A total of 81.9% of women had a diagnosis of anal HSIL and/or cancer and one other genital pathology, with the majority of women having either cervical (41.9%) or vulval (44.0%) HSIL as the genital pathology. Moreover, cervical and vulval HSIL represented 39.8% and 32.5% of all first diagnoses, respectively. This is consistent with published data series on women with multizonal disease, which found that 57% of women had two sites of disease, and 51% and 35% of women had cervical and vulval pathology as a first diagnosis [27], respectively. Multizonal disease was defined as the concurrent presence of HSIL or carcinoma at two or more of the following sites: perianus, anal canal, vulva, vagina or cervix [27].

On review of timelines between diagnoses, there was a median lag of 7 years between the first and second diagnoses, with an average lag of 1–2 years between consecutive diagnoses for those with more than two diagnoses. When broken down by pathology, there was a lag of 14 and 11.5 years between cervical HSIL or cancer and anal cancer and 10 and 7.5 years between cervical HSIL or cancer and anal HSIL, respectively. Unsurprisingly, given the anatomical relationship, the average time between vulval and anal, as well as vaginal and anal pathology, was shorter, with a lag of 6 years between vulval HSIL or cancer and anal cancer, 3 and 1.5 years between vulval HSIL or cancer and anal HSIL, 3.5 and 1.5 years between vaginal HSIL or cancer and anal cancer and 1.5 and 1 year between vaginal cancer or HSIL and anal HSIL, respectively. Suk et al. [28] also investigated similar timelines using SEER data and reported comparable median lags of 13.8, 6.3 and 8.3 years between cervical, vaginal and vulval cancer and a secondary anal cancer diagnosis.

The age at diagnosis in this study was compared with that published in our paper analysing Cancer Outcomes and Services Data Set (COSD) data on all women with anogenital cancers [13], the majority of whom have one site disease. Interestingly, whilst the distribution of age at diagnosis for cervical cancer remained similar between the studies, the age at which women were diagnosed with anal, vaginal and vulval cancer was younger in this study. More specifically, 22.3% vs. 2.5%, 21.1% vs. 3.3% and 33.6% vs. 2.7% of women were diagnosed with anal, vaginal and vulval cancer in the <40 years age group, respectively. Accepting the limitations of directly comparing the two datasets, this is an important observation, which suggests that women with multizonal disease (i.e., the cohort captured in this study) develop anogenital cancers at a younger age than those with single site disease. In fact, in this study 47% of all diagnoses were made in the <40 years age group, with only 6% occurring in the 75+ years age group.

With respect to ethnicity and deprivation, 93.4% of women were of White ethnicity and 55.7% from the most deprived quintiles; this was similar to our study looking at COSD data [13]. High deprivation and ethnic minority populations with HPV-related cancers have been shown to have comparatively poorer prognoses [5,15,29,30,31,32]. Cultural attitudes and misconceptions towards HPV disease, together with low finances and education, all interact and represent significant barriers to healthcare. This consequently results in delayed presentation, more advanced disease at diagnosis and poorer prognosis. In keeping with this, this study found women of Black ethnicity to be significantly more likely to present as an emergency than via the two-week wait pathway (*p* = 0.02).

Lastly, we conducted multivariate multinomial logistic regression analysis to determine the predictors of anal cancer staging as well as number of anogenital diagnoses. Of particular interest were the results comparing the effect of having a cervical vs. a vulval diagnosis on the staging of the anal pathology and the number of subsequent anogenital lesions diagnosed. Compared to patients who were first diagnosed with cervical HSIL, those first diagnosed with vulval HSIL (*p* = 0.05) and cancer (*p* = 0.02) were at decreased odds of having a late-stage anal cancer diagnosis. Similarly, women with a first diagnosis of vulval HSIL were at increased odds (*p* = 0.03) of having an anal HSIL vs. an early cancer diagnosis. This is very interesting, as the incidence of anal cancer has been shown to be greater in women with vulval disease [7]. Consequently, women with vulval disease have been classed in a higher risk category for anal cancer than women with cervical disease [10]. Whilst the data in this study does not dispute this, what it does seem to suggest is that, whilst anal cancer may be more common in women with vulval disease, women with cervical disease are at higher risk of presenting with late-stage anal disease than women with vulval disease. Also, interestingly, women who were first diagnosed with vulval HSIL were found to be at significantly increased odds of presenting with 4–5 vs. 2–3 diagnoses than women first diagnosed with cervical HSIL (*p* = 0.04). This appears to suggest that women who first present with cervical pathology have a lower anogenital lesion burden over time but are at increased risk of presenting with late-stage anal disease compared to women with vulval disease. This is probably linked to the shorter lag time between vulval and anal lesions, which means that anal lesions are identified early whilst patients are receiving treatment for their vulval pathology; with a time lag of 14 years between cervical HSIL and anal cancer, women who are first diagnosed with cervical HSIL are simply lost to any follow-up.

## 5. Strengths and Limitations

There are important limitations associated with both the dataset and its analysis, which need to be acknowledged. To begin with, the data request did not include relevant variables such as smoking, HIV status, number of sexual partners and immunosuppression, as well as data on women with single site anogenital disease, i.e., with only anal cancer or HSIL. This limited our risk factor and regression analyses. The treatment data also lacked clarity with respect to the intention and indication of the treatment.

We were unable to request lifetime data and therefore had to specify the time-period within which a patient could have had a genital and anal diagnosis. We requested data 20 years prior to and 1 year after the anal diagnosis based on the literature, which indicated a lag of 10 to 15 years between cervical and anal pathology [33,34,35,36]. Whilst this was likely sufficient to capture vulval and vaginal pathology, which present at similar ages to anal HSIL and cancer, patients with metachronous cervical lesions could have been missed. Additionally, requesting the data in this manner skews the dataset by only capturing the 20-year period prior to the anal diagnosis; thus, average timelines between genital and anal diagnoses need to be interpreted with caution.

With respect to the analysis of the data, we saw that many of our comparisons on multivariate regression were not interpretable or unreliable. One of the reasons was sparse data or low event counts in some of the categories. One option for correcting this would have been to further combine categories to increase the counts in each group. However, we chose not to do this, as it would have reduced the interpretability and meaning of the data. Then, for certain categories, convergence problems were encountered in the multinomial model, making the corresponding estimates not interpretable (huge estimates, artificially narrow confidence intervals).

Lastly, there are limitations around the definition and coding of anal HSIL and cancer, which need to be taken into consideration. For one, the definition of ‘anal cancer’ changed in 2018 with the AJCC 8th Edition. Before 2018, perianal cancers were classified under the ICD-10 code 44.5, which coded for skin cancers of the trunk. These cancers will therefore not have been included under the ICD-10 21.0 anal cancer code and would have been excluded from this database before 2018. There are also significant issues around HSIL coding, which likely underestimated the number of women with anal and genital HSIL. For one, there is only one code for anal HSIL, D0.13, which is defined as ‘Carcinoma in situ of anus and anal canal (AIN3)’. This represents out of date in nomenclature and does not include p16 positive AIN2. This is also true for the other genital HSILs, as, whilst there are codes for CIN2, VIN2 and VAIN2, they do not discriminate between p16 positive and negative disease, and therefore low-grade squamous intraepithelial lesions (LSIL) and HSIL. To add to the confusion, codes for CIN2, VIN2 and VAIN2 are grouped under ‘Dysplasia’ rather than ‘Carcinoma in Situ’ by the ICD and are excluded from any NCRAS database. The cancer registry also does not collect data on the HPV status or typing.

This emphasises the need for the Multinational Anal Squamous Cell Carcinoma Registry and Audit (mASCARA) [37] database, which will help with the collection of relevant and accurate data on patients with anogenital HSIL and cancer, in turn addressing a lot of the limitations we encountered with the data collected in this NHS Digital data request.

## 6. Conclusions

This study found that 8% of all women in England with anal HSIL and/or cancer had a genital HSIL and/or cancer diagnosis. These proportions were lower than anticipated and were likely influenced by the discussed limitations of Cancer Registry data. Moreover, although it is known that women with genital dysplasia are at higher risk of anal cancer, in the absence of screening, asymptomatic anal HSIL or early cancer is likely underdiagnosed.

Overall, these data demonstrate that cervical and vulval HSIL and/or cancer are important risk factors for anal HSIL and cancer, and, whilst the incidence of anal cancer may be higher in patients with vulval disease compared with cervical disease [7], the occurrence of these conditions in patients with anal cancer and/or HSIL is, in fact, very similar. More importantly, these data suggest that women who are first diagnosed with cervical HSIL have a lower burden of recurrent anogenital lesions over time but are in fact at significantly higher risk of presenting with late-stage anal cancer than women first presenting with vulval pathology. This is likely due to the shorter time lag between vulval and anal pathologies, resulting in anal lesions being picked up during treatment for the vulval condition—something that is less feasible for patients with cervical pathology due the time lag of 14 years between cervical and anal disease. The IANS guidelines currently classify women with cervical pathology in Risk Category B given the lower incidence of anal cancer in this population compared to women with vulval disease, who are in Risk Category A [10]. However, with increased screening of women with cervical disease, we may find the incidence of anal HSIL and early cancer to increase significantly in this patient population, which should arguably be moved to the Risk A category.

Interestingly, the age at diagnosis for anal, vulval and vaginal pathologies was 10–20 years lower for women with multizonal disease compared to women with single site disease [13]. These data suggest that anal cancer screening for this population should start before the age of 50 years or at the very latest 14 or 6 years after cervical or vulval HSIL diagnosis, respectively. This is in line with the current IANS recommendations for anal cancer screening [10], which recommend screening women over 45 years with cervical HSIL or 1-year post vulval HSIL diagnosis.

This study adds to the body of evidence demonstrating the risk of anal cancer in women who had cervical, vaginal or vulval HPV pathology and strengthens the argument to consider anal cancer screening for these patients.

## Figures and Tables

**Figure 1 cancers-17-03970-f001:**
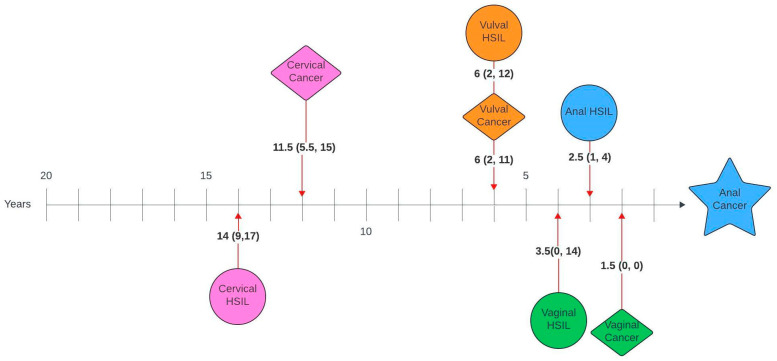
Timelines between genital pathology and anal cancer in women with anal cancer. Median number of years (interquartile range).

**Figure 2 cancers-17-03970-f002:**
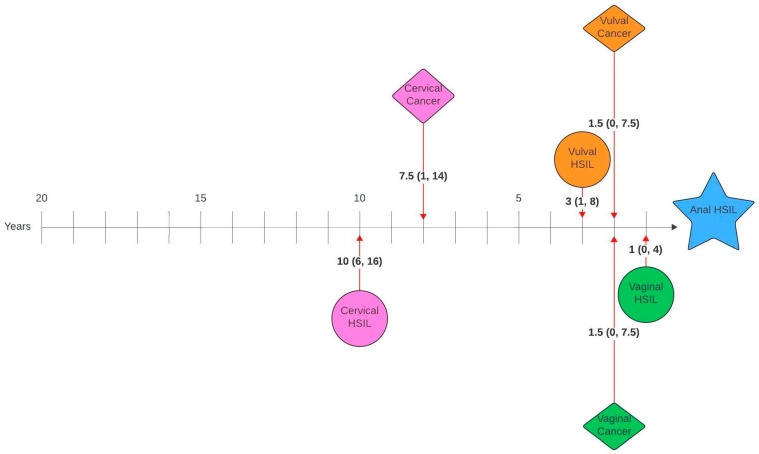
Timelines between genital pathology and anal HSIL in women with anal HSIL. Median number of years (interquartile range).

**Figure 3 cancers-17-03970-f003:**
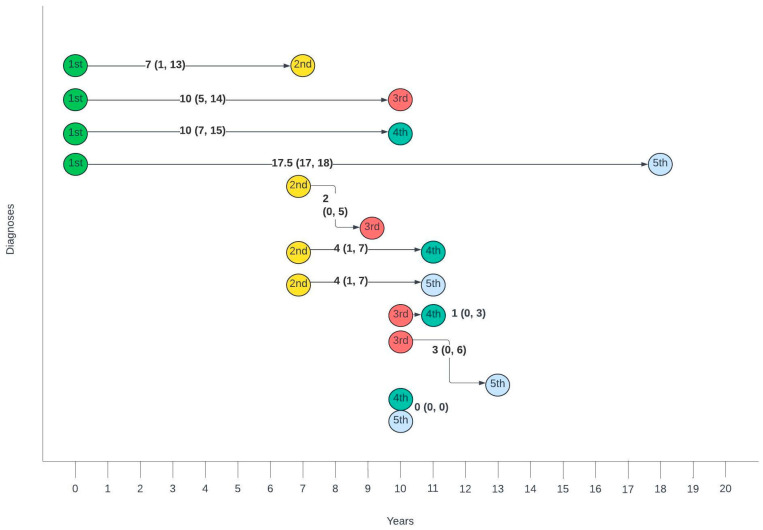
Timelines between consecutive anogenital diagnoses in this cohort. Median number of years (interquartile range).

**Table 1 cancers-17-03970-t001:** Number and distribution of genital diagnoses in patients with anal disease.

	Genital Pathology	Cervical Cancer	Cervical HSIL	Vaginal Cancer	Vaginal HSIL	Vulval Cancer	Vulval HSIL	Anal Cancer	Anal HSIL
Total Pathology (*n* = 2872)	-	72	544	19	52	277	573	734	601
Total Patients	-	72	543	19	52	275	571	733	564
Anal HSIL	564	14	191	2	30	109	376	0	564
% of patients in the study (*n* = 564)	100.0	2.5	33.9	0.4	5.3	19.3	66.7	0.0	100.0
% of patients on a national level (*n* = 3646)	15.5	0.4	5.2	0.1	0.8	3.0	10.3	0.0	15.5
Anal Cancer	733	58	352	17	22	166	195	733	34
% of patients in the study (*n* = 733)	100.0	7.9	48.0	2.3	3.0	22.6	26.6	100.0	4.6
% of patients on a national level (*n* = 12,655)	5.8	0.5	2.8	0.1	0.2	1.3	1.5	5.8	0.3
All Anal Pathology	1297	72	543	19	52	275	571	733	598
% of patients in the study (*n* = 1297)	100.0	5.6	41.9	1.5	4.0	21.2	44.0	56.5	46.1
% of patients on a national level (*n* = 16,301)	8.0	0.4	3.3	0.1	0.3	1.7	3.5	4.5	3.7

**Table 2 cancers-17-03970-t002:** Order of anogenital diagnosis.

Diagnoses	Cervical Cancer (%)	Cervical HSIL (%)	Vaginal Cancer (%)	Vaginal HSIL (%)	Vulval Cancer (%)	Vulval HSIL (%)	Anal Cancer (%)	Anal HSIL (%)	Total (%)
1st	65 (5.0)	516 (39.8)	11 (0.8)	25 (1.9)	188 (14.5)	421 (32.5)	33 (2.5)	38 (2.9)	1297 (100)
2nd	6 (0.5)	25 (1.9)	7 (0.5)	20 (1.5)	64 (4.9)	138 (10.6)	616 (47.5)	421 (32.5)	1297 (100)
3rd	0 (0.0)	3 (1.3)	1 (0.4)	6 (2.6)	20 (8.5)	14 (6.0)	67 (28.5)	124 (52.8)	235 (100)
4th	1 (2.4)	0 (0.0)	0 (0.0)	1 (2.4)	5 (12.2)	0 (0.0)	17 (41.5)	17 (41.5)	41 (100)
5th	0 (0.0)	0 (0.0)	0 (0.0)	0 (0.0)	0 (0.0)	0 (0.0)	1 (50.0)	1 (50.0)	2(100)

**Table 3 cancers-17-03970-t003:** Number of patients in each deprivation quintile related to ethnicity group, number of anogenital diagnoses and route to diagnosis.

Deprivation Quintiles						
	1	2	3	4	5	Total (%)
Total (%)	428 (33)	295 (22.7)	244 (18.8)	188 (14.5)	142 (10.9)	1297(100)
Ethnicity (%)						
White	389 (32.1)	276 (22.8)	228 (18.8)	181 (14.9)	137 (11.3)	1211 (100)
Asian	1 (100)	0 (0.0)	0 (0.0)	0 (0.0)	0 (0.0)	1 (100)
Black	24 (51.1)	11 (23.4)	9 (19.1)	2 (4.3)	1 (2.1)	47 (100)
Mixed and Other	1 (14.3)	4 (57.1)	1 (14.3)	1 (14.3)	0 (0.0)	7 (100)
Unknown	13 (41.9)	4 (12.9)	6 (19.4)	4 (12.9)	4 (12.9)	31 (100)
No. of Diagnoses (%)						
2	341 (32.1)	238 (22.4)	197 (18.5)	158 (14.9)	128 (12.1)	1062 (100)
3	71 (36.6)	49 (25.3)	39 (20.1)	26 (13.4)	9 (4.6)	194 (100)
4	15 (38.5)	8 (20.5)	8 (20.5)	3 (7.7)	5 (12.8)	39 (100)
5	1 (50.0)	0 (0.0)	0 (0.0)	1 (50.0)	0 (0.0)	2 (100)
Route to Diagnosis (%)						
Emergency	38 9 (38.4)	19 (19.2)	20 (20.2)	14 (14.1)	8 (8.1)	99 (100)
GP referral	272 (32.1)	183 (21.6)	172 (20.3)	123 (14.5)	97 (11.5)	847 (100)
Inpatient	8 (34.8)	3 (13.0)	9 (39.1)	1 (4.3)	2 (8.7)	23 (100)
Other Outpatient	146 (32.2)	115 (25.4)	91 (20.1)	58 (12.8)	43 (9.5)	453 (100)
Screening	10 (34.5)	4 (13.8)	6 (20.7)	7 (24.1)	2 (6.9)	29 (100)
TWW	57 (30.5)	41 (21.9)	36 (19.3)	25 (13.4)	28 (15.0)	187 (100)
Unknown	429 (34.8)	290 (23.5)	209 (16.9)	183 (14.8)	123 (10.0)	1234 (100)

**Table 4 cancers-17-03970-t004:** Number of patients in each ethnicity group related to number of anogenital diagnoses and route to diagnosis.

Ethnicity						
	White	Asian	Black	Mixed and Other	Unknown	Total (%)
Total (%)	1211 (93.4)	1 (0.1%)	47 (3.6)	7 (0.5)	31 (2.4)	1297 (100)
No. of Diagnoses (%)						
2	995 (93.7)	1 (0.1)	32 (3.0)	7 (0.7)	27 (2.5)	1062 (100)
3	179 (92.3)	0 (0.0)	11 (5.7)	0 (0.0)	4 (2.1)	194 (100)
4	35 (89.7)	0 (0.0)	4 (10.3)	0 (0.0)	0 (0.0)	39 (100)
5	2 (100)	0 (0.0)	0 (0.0)	0 (0.0)	0 (0.0)	2 (100)
Route to Diagnosis (%)						
Emergency	89 (89.9)	0 (0.0)	7 (7.1)	1 (1.0)	2 (2.0)	99 (100)
GP referral	792 (93.5)	0 (0.0)	33 (3.9)	3 (0.4)	19 (2.2)	847 (100)
Inpatient	20 (87.0)	0 (0.0)	2 (8.7)	0 (0.0)	1 (4.3)	23 (100)
Other Outpatient	412 (90.9)	0 (0.0)	26 (5.7)	6 (1.3)	9 (2.0)	453 (100)
Screening	27 (93.1)	0 (0.0)	2 (6.9)	0 (0.0)	0 (0.0)	29 (100)
TWW	184 (98.4)	0 (0.0)	1 (0.5)	0 (0.0)	2 (1.1)	187 (100)
Unknown	1153 (93.4)	2 (0.2)	42 (3.4)	4 (0.3)	33 (2.7)	1234 (100)

**Table 5 cancers-17-03970-t005:** Staging at diagnosis for all anogenital pathologies.

Stage	Anal (%)	Cervical (%)	Vaginal (%)	Vulval (%)	Total (%)
1	92 (6.9)	13 (2.1)	3 (4.2)	66 (7.8)	174 (6.1)
2	93 (7.0)	2 (0.3)	0 (0.0)	5 (0.6)	100 (3.5)
3	136 (10.2)	4 (0.6)	1 (1.4)	4 (0.5)	145 (5.0)
4	34 (2.5)	1 (0.2)	2 (2.8)	1 (0.1)	38 (1.3)
HSIL	601 (45)	544 (88.4)	52 (73.2)	573 (67.4)	1770 (61.6)
Unknown	379 (28.4)	52 (8.4)	13 (18.3)	201 (23.6)	645 (22.5)
Total (%)	1335 (100)	616 (100)	71 (100)	850 (100)	2872 (100)

## Data Availability

Raw data is unavailable due to ethical restrictions.

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
