# Peer review of "Women with HPV-Driven Anal and Genital Disease: Investigating the Patient Cohort in England"

_cancers, 2025, doi:10.3390/cancers17243970_

Round 1

Reviewer 1 Report

Comments and Suggestions for Authors

In the article entitled “Women with HPV-Driven Anal and Genital Disease: Investigating the Patient Cohort in England”, the authors address a highly relevant topic; however, the manuscript is presented in a format that requires improvement. The study discusses important aspects of the disease, but the lack of clarity and objectivity in the text hinders proper understanding and evaluation of the results and conclusions presented.

With this in mind, I offer the following suggestions:

1. Results in Tables
The results should include tables that clearly and systematically present the data related to the statistical analyses, including significance values and odds ratios.
Figures may be retained as complementary elements to help visualize specific findings, but they should not replace the tables. Demographic data, frequencies of the different types and sites of lesions, as well as treatment information, should be presented in tabular format.

It is recommended that most of the figures be removed, keeping only those that effectively illustrate relevant and complementary aspects of the data shown in the tables.
In the text, only findings with statistical significance or those representing novel and unique results of the study should be highlighted.

Conclusion
I suggest that the manuscript be resubmitted in a more suitable format, emphasizing the organization of the results and the clarity of the writing, in order to enhance comprehension and allow for a more accurate critical evaluation of the study.

Author Response

Thank you so much for your comments. We really value all comments to make this a better manuscript for the reader. We have converted most of the figures to tabular format and revised the text to highlight only findings with statistical significance or those representing novel and unique results of the study.

Reviewer 2 Report

Comments and Suggestions for Authors

The article title: “Women with HPV-Driven Anal and Genital Disease: Investigating the Patient Cohort in England” was reviewed.
Although very good work was carried out but the following questions remain to be answered.
The abstract and the introduction were well-described, but why distribution of HrHPV types has not been mentioned among the patients with anal HSIL/cancer, vulval and/or vaginal and/or cervical cancer HSIL/cancer development.
In the Materials and methods: why, the detection of HrHPV types by molecular techniques was not designed among the patients with anal HSIL/cancer, vulval and/or vaginal and/or cervical cancer HSIL/cancer.
In the results section: why the dominance of HrHPV types was not evaluated among the patients with anal HSIL/cancer, vulval and/or vaginal and/or cervical cancer HSIL/cancer.
Does the implementation HPV vaccination program reduce anal HSIL/cancer, vulval and/or vaginal and/or cervical cancer HSIL/cancer development in UK?

Author Response

  1. Although very good work was carried out but the following questions remain to be answered.
    The abstract and the introduction were well-described, but why distribution of HrHPV types has not been mentioned among the patients with anal HSIL/cancer, vulval and/or vaginal and/or cervical cancer HSIL/cancer development.

Thank you for your comments. We made the following changes to the introduction to reflect your comment on hrHPV types.

‘Anal squamous cell carcinoma (SCC) is a high-risk Human Papillomavirus (hrHPV) induced malignancy [1,2]. People with recurrent and persistent exposure to oncogenic high-risk strains of HPV (16,18, 31, 33,35, 39, 45, 51, 52, 56, 58 and 59) are at higher risk of developing anogenital  neoplastic lesions; in the context of anal cancer, HPV16 is thought to be the most carcinogenic [3].  The incidence of anal cancer has concerningly been on the rise [4,5], and whilst, relatively rare in the general population, affecting 1-2 per 100,000 people[6], there are cohorts of people who are at higher risk, these include people living with HIV (PLWH) (especially men who have sex with men (MSM)), transplant patients and women with other genital HPV driven dysplasia[7].’

  1. In the Materials and methods: why, the detection of HrHPV types by molecular techniques was not designed among the patients with anal HSIL/cancer, vulval and/or vaginal and/or cervical cancer HSIL/cancer.

Thank you for your comment. The cancer registry does not collect information on HPV status or genotyping, so this information was simply not available for request. I have added a comment in the limitations section of the manuscript to reflect this:

‘The cancer registry also does not collect data on the HPV status or typing of the pathology.’

  1. In the results section: why the dominance of HrHPV types was not evaluated among the patients with anal HSIL/cancer, vulval and/or vaginal and/or cervical cancer HSIL/cancer.

Thank you for your comment. The cancer registry does not collect information on HPV status or genotyping, so this information was simply not available for request. I have added a comment in the limitations section of the manuscript to reflect this:

‘The cancer registry also does not collect data on the HPV status or typing of the pathology.’

  1. Does the implementation HPV vaccination program reduce anal HSIL/cancer, vulval and/or vaginal and/or cervical cancer HSIL/cancer development in UK?

Thank you for your comment. The role of the HPV vaccine in the primary prevention of cervical dysplasia is unquestionable. The introduction of the HPV immunisation program in England in 2008 has been shown to reduce the risk of developing cervical HSIL by 97% in those patients vaccinated at the age of 12-13 years; potentially eradicating cervical cancer in women born after the 1st September

1995[1]. It is expected decrease the incidence all anogenital HPV driven pathology, however the extent of this won’t be clear until at least 2050, given that vaginal, vulval and anal dysplasia affect women later on in life at the age of at least 50 years.

[1] Falcaro M, Casta.on A, Ndlela B, et al. The effects of the national HPV vaccination

programme in England, UK, on cervical cancer and grade 3 cervical intraepithelial

neoplasia incidence: a register-based observational study. The Lancet 2021;398(10316):2084-92.

Reviewer 3 Report

Comments and Suggestions for Authors

Very interesting project but needs lots of editing. The way the findings are presented in the text  must be rewritten. I suggest to get a professional editor given that your findings are interesting and you want the reader to understand what you wrote, examples: Pages 9-11, and Conclusions. 
As a rule of thumb, when writing a manuscript you should decide how to best present your data either using tables, graphics, figures, and etc. if you describe your findings in the text you don’t have to repeat in tables or whatever other ways you chose, and vice verse.. Decide which one is best and easy to the reader to follow and get a picture of the results. 
I admit, that in some areas it was very confusing to follow what you wrote.

Thanks for acknowledging the limitations of your work. I wonder if there was any information on sexual behavior among this cohort given the natural history of these infections. For instance, given how the original data is collected, region, SES, and etc can you infer if these women were sexual workers, bisexual, housewives? At least it could be added to limitations of the study. 

Comments on the Quality of English Language

Unfortunately the way the data is presented makes it very difficult to understand and follow. In some instances I had to read several times to get their point. I even got a headache. Pages 9-11 must be rewritten. They authors should decide whether it’s better to just use tables, and graphics  to illustrate their findings, and avoid the text, unless there is a specific reason why they need both. I think, a professional editor should be able to address these  issues..

Author Response

  1. Very interesting project but needs lots of editing. The way the findings are presented in the text must be rewritten. I suggest to get a professional editor given that your findings are interesting and you want the reader to understand what you wrote, examples: Pages 9-11, and Conclusions. 
    As a rule of thumb, when writing a manuscript you should decide how to best present your data either using tables, graphics, figures, and etc. if you describe your findings in the text you don’t have to repeat in tables or whatever other ways you chose, and vice verse.. Decide which one is best and easy to the reader to follow and get a picture of the results. 
    I admit, that in some areas it was very confusing to follow what you wrote.

Thank you so much for your comments. We really value all comments to make this a better manuscript for the reader. We have converted most of the figures to tabular format and revised the text to highlight only findings with statistical significance or those representing novel and unique results of the study. We hope that these changes improve the flow of the manuscript.

  1. Thanks for acknowledging the limitations of your work. I wonder if there was any information on sexual behavior among this cohort given the natural history of these infections. For instance, given how the original data is collected, region, SES, and etc can you infer if these women were sexual workers, bisexual, housewives? At least it could be added to limitations of the study. 

Thank you for your comment. Unfortunately, this information is simply not available from the data request. I have included the following in the limitations section to reflect this point:

‘There are important limitations associated with both the dataset and its analysis which need to be acknowledged. To begin with, the data request did not include relevant variables such as smoking, HIV status, number of sexual partners and immunosuppression as well as data on women with single site anogenital disease, i.e. with only anal cancer or HSIL. This limited our risk factor and regression analyses. ‘

  1. Unfortunately the way the data is presented makes it very difficult to understand and follow. In some instances I had to read several times to get their point. I even got a headache. Pages 9-11 must be rewritten. They authors should decide whether it’s better to just use tables, and graphics  to illustrate their findings, and avoid the text, unless there is a specific reason why they need both. I think, a professional editor should be able to address these  issues.

We have converted most of the figures to tabular format and revised the text to highlight only findings with statistical significance or those representing novel and unique results of the study. We hope that these changes improve the flow of the manuscript

Round 2

Reviewer 3 Report

Comments and Suggestions for Authors

Dear authors thanks for trying to address my concerns. Your study is an important one and should be of great interest to support preventive interventions among this population. However, the way it’s written makes it difficult to follow. There are terms which need to be revised, I give you few examples: genital precancer ? It should be “precancerous genitalia lesions” or “precancerous genital lesions”. Instead of patient group you should used patient population, and when you referred to Cohorts it’s understood to be of patients, you don’t have to keep repeating cohort of patients. As an example, read lane 27. You could say, “ However, this population is poorly researched. 
Lane 40 needs to be rewritten.. what do you mean by “ multizonal disease?”  Maybe you meant “multiple sites involvement”, instead of 10-20 year  less .. it should read “ the age at diagnosis for anal, vulgar and vaginal disease was 10-20 year lower for women with multiple sites when compared to women with single site involvement. Lane 111 “ and severe severity” it should read “ CCI  is a measure of comorbidity severity as follows: mild (1-2), moderate (3-4), and severe (5). Just make it simple for the reader to understand and follow what you did. 

Ask yourself how much of the data you have collected in the tables are of importance to meet your objectives. Also, just give a number and a title to the tables. You don’t need to write “ Table 7.  Table illustrating the first …. . Should read: Table 7.  Initial treatments ( or Baseline treatments) for ano-genital lesions. Table 8. Second treatments for ano-genital lesions ( were these treatments for recurring disease? ) not clear, maybe you need to re-think how to present this findings.

consider : initial, and follow up treatments for recurring ano-vaginal disease. 

Comments on the Quality of English Language

I suggest that you hired a professional editor, given that the study findings is worth publishing. 

Author Response

Dear authors thanks for trying to address my concerns. Your study is an important one and should be of great interest to support preventive interventions among this population. However, the way it’s written makes it difficult to follow.

  1. There are terms which need to be revised, I give you few examples: genital precancer? It should be “precancerous genitalia lesions” or “precancerous genital lesions”.

Thank you for your comment. This term was used in the lay summary. We have now revised this:

‘Anal cancer is more common in women with a background of genital precancerous and cancerous lesions.’

  1. Instead of patient group you should used patient population, and when you referred to Cohorts it’s understood to be of patients, you don’t have to keep repeating cohort of patients. As an example, read lane 27. You could say, “ However, this population is poorly researched. 

Thank you for your comments. These changes have been made.

  1. Lane 40 needs to be rewritten. what do you mean by “ multizonal disease?”  Maybe you meant “multiple sites involvement”,

Thank you, multizonal disease is an accepted term used to describe the presence of high-grade squamous intraepithelial lesions (HSIL)/carcinoma concurrently at two or more of the following sites/zones: perianus, anal canal, vulva, vagina or cervix [1]. We have added the definition of this in the discussion and used the term only where appropriate.

[1] Albuquerque A, Godfrey MAL, Cappello C, Pesola F, Bowring J, Cuming T, De Masi A, Rosenthal AN, Sasieni P, Nathan M. Multizonal anogenital neoplasia in women: a cohort analysis. BMC Cancer. 2021 Mar 6;21(1):232. doi: 10.1186/s12885-021-07949-8. PMID: 33676451; PMCID: PMC7937256.

  1. Instead of 10-20 year  less .. it should read “ the age at diagnosis for anal, vulgar and vaginal disease was 10-20 year lower for women with multiple sites when compared to women with single site involvement.

Thank you, I have edited this.

  1. Lane 111 “ and severe severity” it should read “ CCI  is a measure of comorbidity severity as follows: mild (1-2), moderate (3-4), and severe (5). Just make it simple for the reader to understand and follow what you did.

Thank you I have edited this.

  1. Ask yourself how much of the data you have collected in the tables are of importance to meet your objectives.

Thank you. We have reviewed the tables and would like to include all the current data, as we believe it is relevant to the original data request and aims of the study. We have however, as result of your comments, re-reviewed how we describe the data and made this more succinct.

  1. Also, just give a number and a title to the tables. You don’t need to write “ Table 7.  Table illustrating the first …. . Should read: Table 7.  Initial treatments ( or Baseline treatments) for ano-genital lesions. Table 8. Second treatments for ano-genital lesions ( were these treatments for recurring disease? ) not clear, maybe you need to re-think how to present this findings. consider : initial, and follow up treatments for recurring ano-vaginal disease. 

Thank you, we have amended the table headings. As for the treatment data query, the cancer registry does not specify whether the second treatment is for treatment of the primary disease, for example CRT after a diagnostic surgical biopsy, or persistent disease or recurrent disease. All that is recorded in the cancer registry is the first and second treatment for that specific pathology. The sentence below has been added in the results section to explain this. We also acknowledged this in the limitations section of the manuscript.

‘It is important to note that the intention of the surgical treatment was however not specified, nor was the indication for the secondary treatment i.e. whether it was for definitive treatment, disease persistence or recurrence.’

  1. I suggest that you hired a professional editor, given that the study findings is worth publishing. 

Thank you for your comment. We have thoroughly re-reviewed the manuscript and have made changes to the text to improve the flow of the paper.

Round 3

Reviewer 3 Report

Comments and Suggestions for Authors

Thanks for addressing my concerns. I still encourage you to seek professional assistance from a professional editor. 
Do you need so many tables, figures and graphics? Just try to keep what is really relevant to your most important findings, and delete the rest or just mention it in the text. Please don’t repeat the same information in tables, figures and graphics! Just refer to them in the text. You have turned the manuscript into something quite difficult to read. Just make it simple for the reader!! 

Comments on the Quality of English Language

I suggest that you hired a professional editor, given that the study findings is worth publishing. Too many busy tables. See what you can delete. You have many tables, graphics and figures. Do you really need all of them? Don’t repeat same information across the manuscript unless you are trying to make a point. 
Just choose tables which are relevant. 

Author Response

Thank you for your input. We have now moved 8 of our tables to supplementary materials as per your advice to simplify the manuscript for the reader and avoid repetition.